# Exploring the Regulation of Cytochrome P450 in SH-SY5Y Cells: Implications for the Onset of Neurodegenerative Diseases

**DOI:** 10.3390/ijms25137439

**Published:** 2024-07-06

**Authors:** Alice Pifferi, Elda Chiaino, Jesus Fernandez-Abascal, Aoife C. Bannon, Gavin P. Davey, Maria Frosini, Massimo Valoti

**Affiliations:** 1Dipartimento di Scienze della Vita, Università di Siena, Viale A. Moro 2, 53100 Siena, Italy; alice.pifferi@student.unisi.it (A.P.); elda.ch92@gmail.com (E.C.); aoifecbannon@gmail.com (A.C.B.); massimo.valoti@unisi.it (M.V.); 2Andalusian Centre for Developmental Biology (CABD), CSIC-Universidad Pablo de Olavide-Junta de Andalucía, Carretera de Utrera km 1, 41013 Sevilla, Spain; jferaba@upo.es; 3Department of Molecular Biology and Biochemical Engineering, Universidad Pablo de Olavide, Carretera de Utrera km 1, 41013 Seville, Spain; 4School of Biochemistry and Immunology, Trinity College Dublin, 3533645 Dublin, Ireland; gdavey@tcd.ie

**Keywords:** brain cytochrome P450, neurodegenerative disease, SH-SY5Y cells, drug metabolism, CYP induction

## Abstract

Human individual differences in brain cytochrome P450 (CYP) metabolism, including induction, inhibition, and genetic variation, may influence brain sensitivity to neurotoxins and thus participate in the onset of neurodegenerative diseases. The aim of this study was to explore the modulation of CYPs in neuronal cells. The experimental approach was focused on differentiating human neuroblastoma SH-SY5Y cells into a phenotype resembling mature dopamine neurons and investigating the effects of specific CYP isoform induction. The results demonstrated that the differentiation protocols using retinoic acid followed by phorbol esters or brain-derived neurotrophic factor successfully generated SH-SY5Y cells with morphological neuronal characteristics and increased neuronal markers (NeuN, synaptophysin, β-tubulin III, and MAO-B). qRT-PCR and Western blot analysis showed that expression of the CYP 1A1, 3A4, 2D6, and 2E1 isoforms was detectable in undifferentiated cells, with subsequent increases in CYP 2E1, 2D6, and 1A1 following differentiation. Further increases in the 1A1, 2D6, and 2E1 isoforms following β-naphthoflavone treatment and 1A1 and 2D6 isoforms following ethanol treatment were evident. These results demonstrate that CYP isoforms can be modulated in SH-SY5Y cells and suggest their potential as an experimental model to investigate the role of CYPs in neuronal processes involved in the development of neurodegenerative diseases.

## 1. Introduction

The study of xenobiotic metabolism in the human brain has become increasingly significant, with attention drawn to its potential implications. This goes beyond its conceivable central role in the pharmacological effects of drugs targeting the central nervous system, particularly highlighting its relevance in xenobiotic-induced neurotoxicity [1,2]. Cytochrome P450 (CYP) enzyme content in the human brain varies from 0.5 to 3% compared to that of liver content [3], and they are found in both the endoplasmic reticulum and the mitochondrial inner membrane, while in the liver, they are primarily located in the endoplasmic reticulum [4,5]. CYP enzyme distribution is not uniform among brain regions, and sex-related differences have also been reported [6,7,8]. Therefore, significant disparities exist in their expression across specific cerebral areas, and although their activity in situ is relatively low at the cellular level, the expression can escalate to amounts comparable to those found in hepatocytes, thus potentially exerting a substantial impact on the levels of centrally acting drugs within the brain and, consequently, on their therapeutic efficacy [6]. Furthermore, the impact of CYP enzymes in the brain on the metabolism of catecholamines, cholesterol, and neurotoxic compounds serves as compelling evidence for their potential role in the development of neurodegenerative and psychiatric disorders such as Parkinson’s disease (PD), Alzheimer’s disease, depression, and schizophrenia [7,9,10].

Considering the impact of various brain CYP isoforms on drug/toxin sensitivity and response, it is timely to elucidate whether well-known CYP inducers affect the expression of specific CYP isoforms in the brain. In this regard, a previous study by our research group demonstrated that the cytotoxic effects of 1-methyl-4-phenylpyridinium (MPP^+^, a neurotoxin associated with PD) on undifferentiated (UD) neuroblastoma SH-SY5Y were reverted by the treatment with some CYP inducers, namely, β-naphthoflavone (βNF) and ethanol (EtOH) [11]. To further investigate the impact of CYP inducers on the expression of specific CYP isoforms in the brain, appropriate experimental models are needed. The SH-SY5Y cell line is widely used owing to its ability to undergo differentiation into neuron-like cells [12] by using protocols based on the treatment with retinoic acid (RA), often in the absence or with minimal fetal bovine serum [13]. However, the duration of differentiation after exposure to the treatment can vary significantly in the literature [14], resulting in a population of cells with different characteristics. 

The aim of this study was to investigate the modulation of CYPs in neuronal cells to gain insights into its involvement in the development of neurodegenerative disorders. A suitable cellular experimental model based on differentiation of human neuroblastoma SH-SY5Y cells was set up by means of two different protocols using retinoic acid (RA) along with a phorbol esters such as 12-O-Tetradecanoylphorbol-13-acetate (TPA) or the brain-derived neurotrophic factor (BDNF), with the final aim to obtain a dopaminergic-like mature neuron phenotype. In differentiated cells, after performing an appropriate morphological analysis as well as assessing the expressions of different generic (neuronal nuclear protein, NeuN, synaptophysin, and β-Tubulin III) and dopaminergic-specific (dopamine transporter, monoamine oxidase A and B) markers of mature neurons, the basal mRNA and protein expression of several CYP isoforms (CYP1A1, CYP2B6, CYP2D6, CYP2E1, and CYP3A4) was performed. Moreover, the possibility to induce some CYPs upon the treatment with well-known inducers such as βNF and EtOH was explored in terms of their putative mRNA levels and protein expression.

## 2. Results

### 2.1. Effect of Differentiation with RA-TPA and RA-BDNF on SH-SY5Y: Morphological Analysis 

Images of SH-SY5Y cells were captured using a phase contrast microscope, and a visual analysis was conducted to assess morphological changes. Undifferentiated SH-SY5Y cells (UD) displayed a large, flat, epithelial-like phenotype, tending to grow in clusters with short, truncated processes extending from cells at the cluster edges (Figure 1). 

As the cells underwent differentiation, proliferation rate decreased, and they spread more evenly throughout the flask. Over time, neurite outgrowth, which is the process by which neurons develop axons and dendrites, became apparent, accompanied by branching and the formation of detectable networks. This was particularly evident in the RA-BDNF differentiation protocol at day 7. The differentiated cells exhibited a morphology resembling neurons, with more polarized cell bodies, and there was a significant increase in neurite length in both the RA-TPA (1.9-fold) and RA-BDNF (2.4-fold) protocols compared to undifferentiated SH-SY5Y cells.

### 2.2. Expression of Specific Neuronal Markers

To further investigate the development of neuronal features during differentiation, the expression of neuron—specific markers, namely, β-tubulin III, synaptophysin, neuronal nuclear protein (NeuN), and the dopamine transporter DAT, which serves as a specific marker for dopaminergic neurons, were analyzed. Protein levels of these neuronal markers were compared between undifferentiated (UD) and differentiated SH-SY5Y cells using Western blot analysis. As shown in Figure 2, both RA-TPA- and RA-BDNF-differentiated cells exhibited increased expression in all the tested neuronal markers. Among them, synaptophysin and β-tubulin III showed the most significant changes, with an approximately threefold and fourfold increase, respectively, in both RA-TPA- and RA-BDNF-differentiated cells. Finally, NeuN and DAT were also increased by approximately two- to threefold.

To gain a deeper understanding of the differentiated SH-SY5Y cell populations and to examine the changes in the expression of enzymes involved in dopamine metabolism, the expression of MAO A and MAO B was investigated. As shown in Figure 3, the differentiation protocol had no impact on the expression of MAO A. However, a significant increase of approximately fourfold in MAO B protein levels was observed in fully differentiated cells treated with both RA-BDNF and RA-TPA protocols.

### 2.3. Effect of Differentiation on Basal CYP450 Expression 

The expression of mRNA and protein levels of several CYP450 isoforms (CYP1A1, CYP2B6, CYP2D6, CYP2E1, and CYP3A4) in both undifferentiated (UD) and differentiated SH-SY5Y cells, was investigated (Figure 4). Differentiation with RA-BDNF had a slight effect on the mRNA levels of CYP1A1, resulting in a 2.2-fold increase. On the other hand, the RA-TPA protocol was more effective, leading to a 5.9-fold increase in CYP1A1 mRNA levels. Interestingly, the changes in CYP2D6 mRNA levels were the opposite, increasing by 4.9-fold in the case of RA-BDNF, while they remained unchanged in the case of RA-TPA. Both protocols had an impact on CYP2E1 mRNA levels, as it increased by 3.8-fold with RA-BDNF and by 9.8-fold with RA-TPA, while the expression of CYP3A4 was mostly unaffected by the differentiation process. Notably, PCR amplification did not show the presence of CYP2B6 isoform in either undifferentiated SH-SY5Y cells or the two neuron-like cell populations obtained with BDNF or TPA.

The expression of CYP isoform proteins was assessed through Western blot immunoassay. As shown in Figure 5, CYP2D6 and CYP2E1 proteins were detectable in UD cells. The differentiation induced by RA-TPA did not influence the expression of these proteins. However, with RA-BDNF differentiation, an increase in CYP2D6 expression was observed, while the expression of CYP2E1 remained unchanged. Finally, both undifferentiated and differentiated cells did not exhibit detectable levels of CYP1A1 and CYP3A4 proteins.

### 2.4. Induction of CYP Isoforms in UD and Differentiated SH-SY5Y Cells 

The possibility to induce CYP3A4-2D6-2E1-1A1 in UD and differentiated SH-SY5Y cells upon the treatment with well—known inducers such as βNF and EtOH was then investigated. For **CYP1A1**, results showed that βNF increased its mRNA content by 1.9-fold in UD cells, while it was less effective in RA-BDNF and in RA-TPA cells, in which differentiation already elevated this isoform by 2.2-fold and 5.9-fold, respectively (Figure 6).

Changes in mRNA 1A1 levels were also observed when EtOH was used, as a 1.8-fold increase occurred in UD cells, while its expression was comparable to that of the respective differentiated cells taken as controls in RA-BDNF and RA-TPA cells. βNF was ineffective in changing CYP3A4 expression in all SH-SY5Y population under investigation. On the other hand, EtOH-mediated effects were similar, although an upward trend seemed to occur in UD and in RA-BDNF. **CYP2D6** mRNA was found to be raised by both inducers in UD cells, in which an increase in its expression of 1.6- and 1.8-fold by βNF and EtOH, respectively, was found. In both RA-BDNF- and RA-TPA-differentiated cells βNF and EtOH seemed ineffective towards this CYP isoform, although an upward trend seemed to occur in the RA-TPA cells upon inducer treatment. Interestingly, however, CYP2D6 mRNA was significantly upregulated by the differentiation per se in the RA-BDNF protocol, but not in that using RA-TPA. In the case of **CYP2E1**, βNF- and EtOH-mediated effects showed a trend in the increase in this isoform in UD cells. In the differentiated cells, a basal increase in 2E1 mRNA expression was triggered by differentiation (RA-BDNF 6.4-fold, *p* < 0.05 vs. CTRL UD; RA-TPA 9.8-fold vs CTRL UD, *p* = 0.061), while with both βNF and EtOH, the mRNA of this isoform was not affected. Finally, qRT-PCR showed no mRNA expression of the 2B6 isoform in all the SH-SY5Y models, and no induction was observed upon βNF and EtOH treatment. 

WB quantitative analysis demonstrated an increase in the expression of CYP2D6 and CYP2E1 by βNF treatment in undifferentiated (UD) cells, whereas EtOH did not have the same effect (Figure 7). Additionally, the RA-BDNF differentiation protocol itself led to an increase in the expression of these specific CYP isoforms, while the RA-TPA protocol appeared to be less effective. In differentiated SH-SY5Y cells, however, the protein expression of CYP2D6 and CYP2E1 remained mostly unchanged following treatment with βNF and EtOH. Furthermore, CYP1A1 and CYP3A4 were not detectable in both undifferentiated and differentiated cells.

## 3. Discussion

The human cell line SH-SY5Y has biochemical characteristics similar to human dopaminergic (DAergic) neurons as they express D1 and D2 receptors, DA synthesis enzymes, as well as DA transporter [15,16]. Their usefulness for research purposes is in the possibility of being differentiated into a mature neuron-like phenotype cell population, but depending on the protocols used, some studies have found that the differentiated SH-SY5Y cell line expresses molecular markers associated with catecholaminergic phenotypes, particularly noradrenergic and dopaminergic markers such as dopamine-β-hydroxylase and tyrosine hydroxylase (TH), while other reports indicate low or negligible expression of dopaminergic markers in this cell line [17]. For these reasons, it is vital to establish and validate a protocol that drive the growth of undifferentiated cells towards a dopaminergic phenotype. For the differentiation, several agents such as RA [18,19], BDNF [20], TPA [15,21], staurosporine [22,23], and conditioned medium of human neural stem cells (CM-hNSCs) [24] have been used alone or in combination [15] to generate a homogenous population of neuronal-like cells. Moreover, reducing the amount of FBS in culture medium to 10% [25] or lower (3–1%) [26,27,28] was reported to facilitate the differentiation process [29].

The RA-mediated differentiation of SH-SY5Y neuroblastoma cells promotes increased expression of choline acetyl transferase (ChAT), vesicular monoamine transporter (VMAT), and NA production, without affecting tyrosine hydroxylase (TH) and dopamine transporter (DAT) [15,26]. With a few exceptions, the evidence reported by several studies suggests that these cells are shifted towards an adrenergic or cholinergic phenotype, and this is why this type of differentiation was not used herein. On the contrary, the differentiation protocol consisting in the treatment with RA followed by TPA or by BDNF leads SH-SY5Y cells to develop a DAergic phenotype with a high expression of D2 and D3 receptors, and increased expression of TH and DAT [20,30]. Accordingly, the protocols used in the present study, i.e., RA-TPA and RA-BDNF, resulted in efficient differentiation, driving structural transformation of undifferentiated SH-SY5Y cells to mature neuronal-like cells characterized by branched or unbranched neurite outgrowth and elongated cell bodies, connection to the adjacent cells, reduced proliferation, and stability of the cell population. 

RA-based differentiation of SH-SY5Y cells induces TrkB receptor expression, and their activation is induced by neurotrophins, like BDNF, triggering morphological changes towards neuronal phenotype [31]. Compared to cells treated with RA alone, in fact, fully differentiated cells (RA-BDNF) formed extended neurites, which were accompanied by a low proliferation rate and, importantly, the expression of mature neuronal markers. Similarly, the cells treated first with RA and subsequently with TPA behaved similarly in terms of cell growth and morphological changes induced by differentiation, as well as in the expression of mature neuronal markers, which were comparable to those found in the RA-BDNF population. An increase in NeuN expression was in fact observed in cells differentiated with both protocols used (RA- BDNF and RA-TPA), while RA-treated cells were negative for NeuN, as also found elsewhere [12]. Differentiation of SH-SY5Y cells also results in changes in the expression of some genes encoding for neuronal markers, including synaptophysin [23,32]. The current results confirmed these data by showing how a significant increase in synaptophysin expression occurred in each cell population tested, particularly in RA-BDNF-differentiated SH-SY5Y. Moreover, the differentiation significantly elevates the expression of DAT [15] and a comparable expression of this transporter was also shown in RA-TPA and RA-BDNF differentiated SH-SY5Y cells in the present study, suggesting that both these methods equally drive cells towards a DAergic neuronal phenotype.

The increased in MAO B levels found in differentiated cells deserves a special comment. As is well known, MAO B is primarily localized in astrocytes whereas MAO A is largely found in neurons [33]. Neurodegenerative diseases such as PD and AD are characterized by an increase in MAO activity enzymes, although the understanding of whether this is an ancillary process or a contribution to neurons loss is still under debate [34]. In transgenic adult mice in which the astrocyte expression of MAO-B was selectively induced, a specific, and gradual degeneration of substantia nigra dopaminergic neurons mirroring changes occurring in PD was reported [35]. These changes correlated with decreased locomotor activity and suggested that MAO-B could be directly involved in multiple aspects of the pathology of neurodegenerative diseases. The evidence that differentiation can modulate several proteins and enzymes involved in the metabolism of neurotoxins justifies the use of this model in the study of the neurodegeneration promoted by exogenous as well endogenous compounds (i.e., salsolinol).

CYP enzymes metabolize endogenous neuroactive substances in the brain, while the brain is in turn engaged in the central neuroendocrine and neuroimmune regulation of CYP in the liver. This kind of bifunctional relationship between the brain and cytochrome P450 exert a direct impact on a wide range of processes, with confirmed effects on many centrally acting drugs response [36]. On the other hand, since drugs acting on the CNS have the potential to affect cytochrome P450 in both liver and brain, they may affect their effects at both a pharmacokinetic and pharmacodynamic level. The presence and regulation of CYP content in the brain have been previously demonstrated by us [5] and other research groups (for a comprehensive review, see [3]). In this context, we demonstrated that brain CYPs can metabolize l-deprenyl to a similar extent in both mice and monkey brain microsomal preparations, a process leading to the formation of methamphetamine, a metabolite involved in the therapeutic effects of the MAOB inhibitor [37]. 

Moreover, in both Neuro-2A and primary dopaminergic neuron cells, Bajpai et al. [38] demonstrated that mitochondria CYP2D6 plays a crucial role in the metabolism of MPTP into its toxic cationic form, MPP^+^, thereby contributing to dopaminergic neuron dysfunction. Additionally, the same research group suggested that CYP2D6 could also be involved in the metabolic activation of β-carboline and isoquinoline derivatives, which are suggested as endogenous toxins implicated in the onset of idiopathic PD [39]. These observations outlined the importance of cell-based models of PD such as those used in the present study for studying the role of neurotoxins and CYP in the onset of this pathology, as they provide a controlled environment crucial to elucidating the cellular mechanisms and molecular interactions involved.

In a previous report, we demonstrated that when treated with either ßNF or EtOH, undifferentiated SH-SY5Y cells displayed an upregulation of CYP mRNA for CYP1A1, 2D6, 2E1, and 3A4 but not 2B6 isozymes, an effect that was accompanied by increased protein levels of CYP2D6 and CYP2E1, as shown by Western blotting analysis [11]. In the current study, we aimed at investigating whether dopaminergic-like SH-SY5Y differentiated cells respond similarly upon treatment with the same CYP inducers. The differentiation protocols, however, already resulted in differential expression of these isoforms, with many cases showing an inherent increase in mRNA CYP isozyme in differentiated vs UD cells. Specifically, RA-BDNF increased the expression of 1A1, 2D6, and 2E1, while RA-TPA treatment led to increased expression of 1A1 and 2E1, leaving 3A4 and 2D6 levels largely unaffected compared to their levels in UD cells. Furthermore, both differentiation treatments appear to induce maximal mRNA expression, as subsequent treatment with ßNF and EtOH did not substantially increase mRNA content, in contrast to what was observed in undifferentiated SH-SY5Y cells.

Quantification of CYP isozymes via Western blot analysis revealed that only CYP2D6 was increased compared to UD cells following both differentiation protocols, while CYP2E1 protein levels remained similar to those in UD cells. Conversely, CYP1A1 and 3A4 were undetectable under the present experimental conditions. However, this was not surprising, as reports indicate that for many CYPs, there is a lack of correlation between mRNA, protein, and enzyme activity. Frequently, in fact, alterations in mRNA do not align with corresponding adjustments in the protein, and vice versa [3,40]. Moreover, according to PCR analysis, CYP2D6 and 2E1 isoforms had already reached their maximum levels following the differentiation processes. Notably, RA-BDNF treatment appeared to be more effective than RA-TPA in enhancing the translation of CYP2D6 and 2E1.

Kloditz et al. [41], in an in vitro system utilizing 2D or 3D human hepatocytes, observed that proinflammatory cytokines exert significant effects on the gene expression of certain drug-metabolizing enzymes, both phase I and phase II, via the activation of kinase signaling cascades. Interestingly, the regulation of the studied CYP genes appeared to be independent of nuclear factors known to govern CYP expression. Moreover, the post-translational expression of drug-metabolizing enzymes appeared to be related to the regulation of various interfering RNAs.

We can speculate that a similar phenomenon may occur during the differentiation process of SH-SY5Y cells. The activation of intracellular kinase signaling induced by BDNF or TPA seemed to result in maximum increases in gene expression of the studied CYPs, independently of the regulation of nuclear factors affected by the CYP inducers. Additionally, in terms of protein abundance, the differentiation process did not appear to be closely linked to mRNA levels. While in some cases, mRNA increased up to 6-fold compared to those in UD cells, upon conducting Western blot analysis, it was evident that the highest protein levels were less than 3.5-fold those present in UD cells (i.e., CYP2D6). Moreover, for other isoforms such as CYP1A1 and CYP3A4, the relative protein levels were undetectable in the Western blot analysis. These findings suggest that post-translational regulation of CYP isoforms can be strongly influenced by some forms of interfering RNAs, in line with the observations of Kloditz and colleagues [41], although other hypothesis cannot be ruled out.

## 4. Materials and Methods

### 4.1. Chemicals

All the reagents were purchased from Sigma Merck (Darmstadt, Germany), except otherwise indicated.

### 4.2. SH-SY5Y Cells: Culture and Differentiation 

Neuroblastoma SH-SY5Y cells (Sigma Merck, Darmstadt, Germany, passage 4–12) authenticated by gene typing (LGC Standards S.r.L., Milan, Italy) were cultured as described by Kovalevich and Langford [42] with some modifications. Undifferentiated SH-SY5Y cells were maintained in Dulbecco’s Modified Eagle’s Medium Low glucose (Euroclone, Pero, Italy), supplemented with 10% heat-inactivated FBS (Euroclone, Pero, Italy), 1% of L-glutamine, and 1% penicillin/streptomycin (100 U/mL and 100 µg/mL), in the presence of 5% CO_2_ in a humidified incubator at 37 °C. The growth medium was refreshed every 2 days, and the cells were subcultured once 80–90% confluence was reached. All solutions for cell culture were heated to 37 °C before use.

Two differentiation protocols were employed based on that used by Forster [18], with some modifications according to the scheme reported in Figure 8. SH-SY5Y cells were seeded as appropriate (1 × 10^5^ for a 6-well-plate; 4 × 10^4^ for a 12-well plate; 1.6 × 10^4^ for a 96-well plate). After 24 h, the medium was aspirated, cells were washed with PBS, and differentiation medium (DM) was added according to the following protocols: (a) DM with retinoic acid (RA) and 12-O-Tetradecanoylphorbol-13-acetate (TPA): Cells were treated with 10 µM RA in 1% FBS differentiation medium. After three days without medium change, cells were washed with PBS, added with 1% FBS differentiation medium containing 80 nM of TPA and maintained under these conditions for a further three days (RA-TPA). (b) DM with RA and Brain-Derived Neurotrophic Factor (BDNF): Cells were treated with 10 µM RA in 1% FBS differentiation medium. After three days without medium change, cells were washed with PBS, added with 1% FBS differentiation medium containing 50 µg/mL of BDNF, and were thus maintained under these conditions for a further three days (RA-BDNF). All treatments were performed under dark conditions, as the compounds used were photosensitive. Differentiation was monitored daily with phase contrast microscopy for neurite outgrowth and slowing of proliferation. On day 7, differentiated SH-SY5Y cells were checked for differentiation markers. 

### 4.3. Morphological Analysis 

During and at the end of differentiation, the cells were observed under a phase-contrast microscope and photos were randomly taken. Morphological changes were quantified by an operator who was blind to the treatments by using ImageJ software (National Institute of Health, Bethesda, MD, USA, 1.37v). A total of 3–5 photos/well were taken and the number and length of neuritic processes were then counted and measured. 

### 4.4. CYP Induction

Induction of CYP3A4-2D6-2E1-1A1 in UD and differentiated SH-SY5Y cells (day 7, see Figure 1) was analyzed by separately treating them with βNF (4 μM, 48 h) or EtOH (100 mM, 48 h) [11,43]. Western blot or quantitative real-time PCR analysis was performed as detailed below. 

### 4.5. Western Blot Analysis

The protocol used was described by Peach et al. [44] with some modifications. The cells were washed twice with cold PBS and scraped by 1X RIPA lysis buffer containing protease and phosphatase inhibitor cocktail (1:100). The suspension was transferred into a new tube, vortexed (30 s) and incubated on ice (~5 min), then centrifuged (13,000× *g*, 15 min at 4 °C), and the resulting supernatant was stored at –80 °C until use. After protein extraction, electrophoresis was performed as already described [11]. Quantities of 30–250 μg/sample were diluted in loading buffer and then heated for 8 min at 95 °C in a Thermo-shaker. Samples and molecular weight markers were then loaded in a 10% gel of polyacrylamide for the electrophoresis stage. For protein separation, the gel was exposed to a current of 400 mA and an electric potential of 135 V, lasting approximately 90 min. After electrophoresis, the transfer was carried out as described by Komatsu [45] with some modifications according to manufacturer indications. The 10% gel of polyacrylamide was assembled in a sandwich with the PVDF card to transfer the separated proteins from the gel. The sandwich was then loaded in a tank containing transfer solution buffer and exposed to an electric potential of 100 V for 75–90 min. The PVDF membrane was incubated in probing blocking solution for 1 h, followed by three washes with probing wash buffer for 5 min each, then incubated overnight with primary antibodies or β-actin as housekeeping protein (Appendix A). The following day, the PVDF membrane was washed three times with probing wash buffer and incubated with secondary antibodies (Appendix A) for 1 h. After incubation, the membrane was washed two times with probing wash buffer and once with buffer w/o Tween. Finally, PVDF cards were scanned for bands and analyzed with the Molecular Imager Gel DocTM XR System. The intensity of each band was calculated with ImageJ software (Bethesda, MD, USA, https://imagej.net/software/imagej/#publication, accessed on 15 January 2024) and normalized with the β-actin.

### 4.6. Quantitative Real-Time PCR 

After treatments of differentiated or undifferentiated SH-SY5Y cells, the RNA was extracted with TRIzol™ reagent according to the manufacturer’s instructions (see Appendix A). RNA amount was determined with the NanoDrop ND-1000 UV-Vis Spectrophotometer (Thermo Fisher Scientific Waltham, MA, USA), using the software ND-1000 V3.3.0 (Thermo Fisher Scientific Waltham, MA, USA). mRNA reverse transcription to cDNA was performed by using the High-Capacity cDNA Reverse Transcription kit according to manufacturer indications (Thermo Fisher Scientific, Waltham, MA, USA) in a 9800 Fast Thermal Cycler from Applied Biosystems (Thermo FisherScientific Waltham, MA, USA). The resulting cDNA was used for a PCR amplification by using a StepOne™ Real-Time PCR System (thermal conditions are reported in Appendix A) and data were acquired with StepOne 2.0 software (Thermo FisherScientific Waltham, MA, USA).

Particularly, SsoAdvancedTM Universal SYBR^®^ Green Supermix (BioRad, Hercules, CA, USA) and specific primers were used to detect human CYP1A1, 2E1, 2D6, 2B6, and 3A4 (see Appendix A). The 2^−ΔΔCt^ method was used to obtain the relative quantization of gene expression and results were shown as fold change values compared with the mean of control untreated samples.

### 4.7. Data Analysis 

Results are reported as mean ± SEM of at least 3 independent experiments and normalized to control values when appropriate. In the case of Western Blot and qRT-PCR, statistical analysis was performed by using a one-sample t-test. For the other assays, one-way ANOVA followed by Bonferroni post-test was used (GraphPad Prism version 6.01 for Windows, GraphPad Software, La Jolla, CA, USA, www.graphpad.com). *p* < 0.05 was considered to be significant.

## 5. Conclusions

The presence and regulation of CYP isozymes in SH-SY5Y cells justify the use of this model for studying the involvement [46] of drug metabolism systems in the therapeutic effects of drugs that impact the brain, as well as in neurodegeneration promoted by exogenous or endogenous compounds. The role of CYP in neurodegenerative processes leading to Parkinson’s disease has gained relevance considering recent research demonstrating that 26 single-nucleotide polymorphisms (SNPs) among the 57 CYP genes in humans are more prevalent, with an OD value greater than fivefold, in Parkinson’s patients compared to healthy individuals [47].

## Figures and Tables

**Figure 1 ijms-25-07439-f001:**
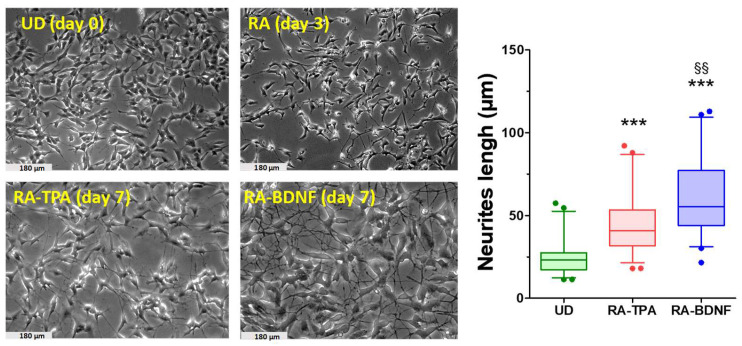
Morphological changes in SH-SY5Y cells promoted by differentiation. The undifferentiated cells (UD) tend to grow in clusters and have a low polarized body with short, truncated neurites. (RA): In the 3-day RA differentiation period, a marked polarization of the cell body occurred, and neurites started to elongate. The differentiation with RA and TPA (RA-TPA) trigger the outgrowth of a neuritic network, and cells maintained a more polarized cell body than in UD. The most striking difference was noted in the BDNF-differentiated protocol (RA-BDNF) as cells had a more rounded cell body, with dense and interconnected neuritic networks (scale bar 180 µm). Boxplot: neurite lengths in SH-SY5Y cells cultured in medium (undifferentiated, UD) or medium supplemented with RA + TPA or RA + BDNF. Box between 25th and 75th percentile; horizontal line: median; whiskers: 5th and 95th percentiles; circles: outliers. Measured neurites: *n* = 54. Statistical analysis was carried out by one-way ANOVA followed by Bonferroni post-hoc test. *** *p* < 0.001 vs. UD cells; §§ *p* < 0.01 vs. RA-TPA cells.

**Figure 2 ijms-25-07439-f002:**
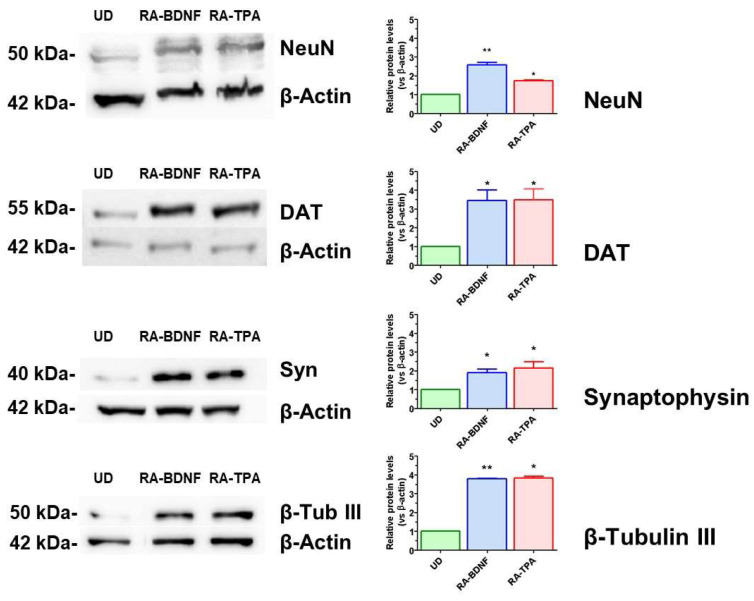
Neuronal protein marker levels in undifferentiated (UD) and differentiated SH-SY5Y cells. WB analysis of undifferentiated (UD) or differentiated (RA-BDNF, RA-TPA) SH-SY5Y cell lysates. The left panels depict a representative blot of proteins of interest in SH-SY5Y cells derived from 30 µg of total cellular lysate, with molecular mass standards (kDa) reported on the left of each WB image. Right panels: quantitative analysis for the expression of proteins NeuN, DAT, synaptophysin, and β-tubulin III. Bars represent the mean ± SEM of 3–4 independent experiments. The values of the treated samples (differentiated cells) were compared to the respective control (undifferentiated cells, UD), which was taken as 1-fold level. All samples were normalized with β-actin as housekeeping protein. Statistical analysis was carried out by one-sample *t*-test. * *p* < 0.05 ** *p* < 0.01 vs. UD samples.

**Figure 3 ijms-25-07439-f003:**
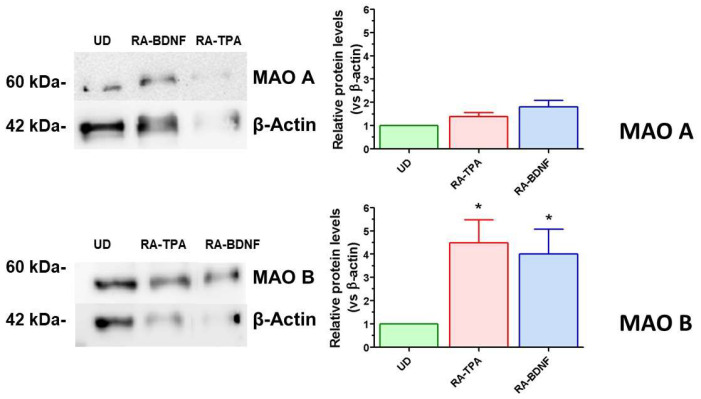
MAO A and MAO B levels in undifferentiated (UD) and differentiated SH-SY5Y cells. The left panel depicts a representative WB of MAO A and MAO B proteins from 30 µg of total SH-SY5Y cellular lysate of undifferentiated (UD) or differentiated (RA-BDNF, RA-TPA) cells. Right panel: WB quantitative analysis. Bars represent the mean ± SEM of 3–4 independent experiments. The values of the treated samples (differentiated cells) were compared to the respective control (undifferentiated cells, UD), which was taken as 1-fold level. All samples were normalized with β-actin as housekeeping protein. Statistical analysis was carried out by one–sample *t*-test. * *p* < 0.05 vs. UD samples.

**Figure 4 ijms-25-07439-f004:**
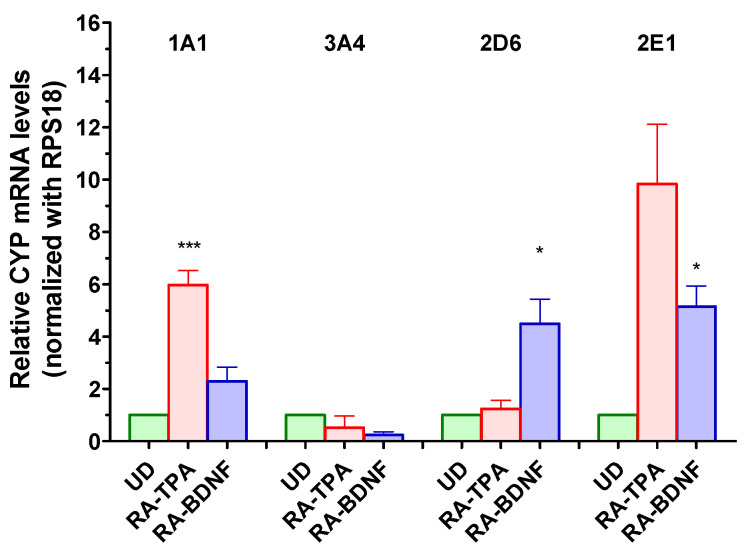
Relative basal mRNA levels of CYP 1A1, 2D6, 2E1, and 3A4 in undifferentiated (UD) and differentiated SH-SY5Y cells. Columns represent the mean ± SEM of 5–8 independent experiments. The values of the differentiated cells were compared to the respective controls (undifferentiated cells, UD), which was taken as 1-fold level. All samples were normalized with RPS18 as housekeeping. Statistical analysis was carried out by one-sample *t*–test. *** *p* < 0.001, * *p* < 0.05 vs. UD CTRL of the respective CYP isoform.

**Figure 5 ijms-25-07439-f005:**
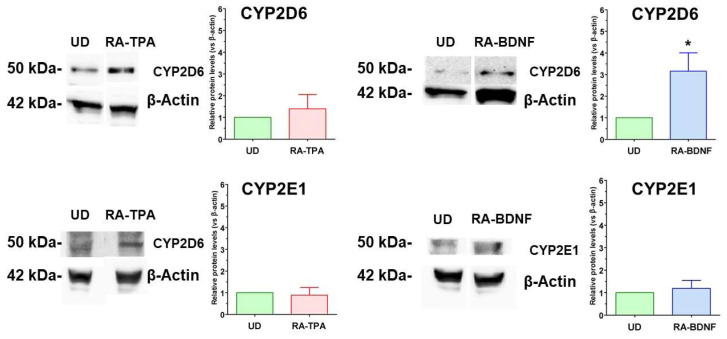
Relative basal levels of CYP 2D6 and 2E1 in undifferentiated (UD) and differentiated SH-SY5Y cells. The left panel depicts a representative WB of CYP2D6 and CYP2E1 proteins from 60–250 µg of total SH-SY5Y cellular lysate of undifferentiated (UD) or differentiated (RA-BDNF, RA-TPA) cells. Right panel: WB quantitative analysis. Bars represent the mean ± SEM of 5–7 independent experiments. The values of the treated samples (differentiated cells) were compared to the respective control (undifferentiated cells, UD), which was taken as 1-fold level. All samples were normalized with β-actin as housekeeping protein. Statistical analysis was carried out by one-sample *t*-test. * *p* < 0.05 vs. UD samples.

**Figure 6 ijms-25-07439-f006:**
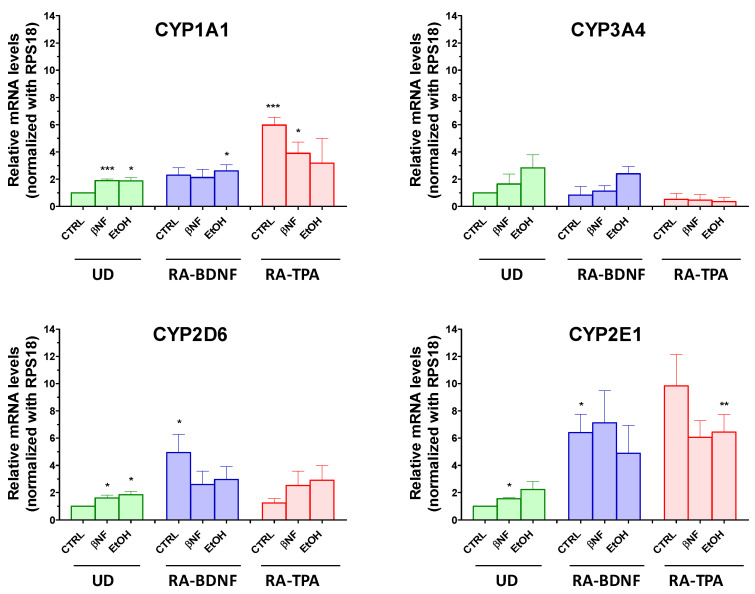
Relative mRNA levels of CYP1A1, CYP3A4, CYP2D6, and CYP2E1 in undifferentiated (UD) or differentiated SH-SY5Y cells following treatment with βNF (4 µM, 48 h) or EtOH (100 mM, 48 h). Columns represent the mean ± SEM of 4–6 independent experiments. The values of the treated samples were compared to the control sample, which was taken as 1-fold level. All samples were normalized with RPS18 as housekeeping. Statistical analysis was carried out by one-sample *t*-test * *p* < 0.05, ** *p* < 0.01, *** vs. *p* < 0.001 vs. UD CTRL.

**Figure 7 ijms-25-07439-f007:**
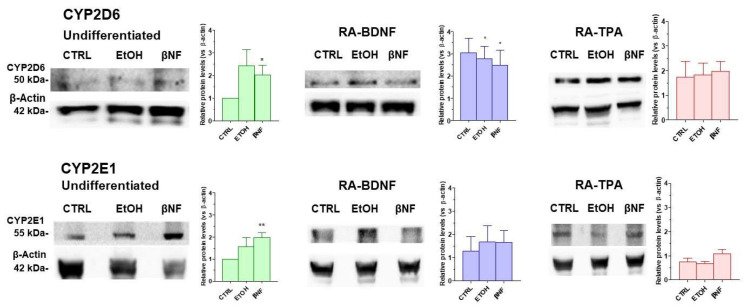
Relative levels of CYP2D6 and CYP2E1 in undifferentiated (UD) or differentiated SH-SY5Y cells following the treatment with EtOH (100 mM, 48 h) or βNF (4 µM, 48 h). The left panel depicts a representative WB of CYP2D6 and CYP2E1 proteins from 60–250 µg of total SH-SY5Y cellular lysate of undifferentiated (UD) or differentiated (RA-BDNF, RA-TPA) cells. Right panel: WB quantitative analysis. Bars represent the mean ± SEM of 5–8 independent experiments. The values of the treated samples (differentiated cells) were compared to the respective control (undifferentiated cells, UD), which was taken as 1-fold level. All samples were normalized with β-actin as housekeeping protein. Statistical analysis was carried out by one-sample t-test. * *p* < 0.05, ** *p* < 0.01 vs. UD samples.

**Figure 8 ijms-25-07439-f008:**
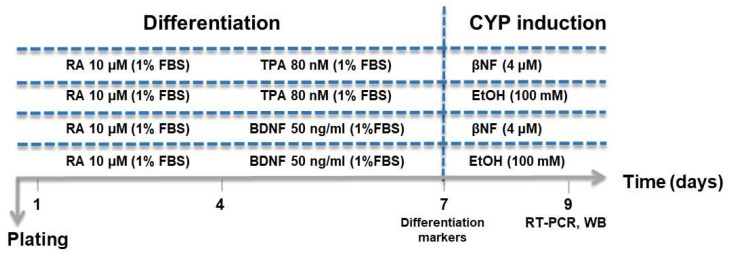
Experimental protocols used for the differentiation of SH-SY5Y cells into DA-like neurons, and induction of CYP3A4-2D6-2E1-1A1.

## Data Availability

Data will be made available on request.

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
