# Peer review of "Exploring the Regulation of Cytochrome P450 in SH-SY5Y Cells: Implications for the Onset of Neurodegenerative Diseases"

_ijms, 2024, doi:10.3390/ijms25137439_

Round 1

Reviewer 1 Report

Comments and Suggestions for Authors

The article of Alice Pifferi et al. “Exploring the regulation of cytochrome P450 in SH-SY5Y cells: implications for the onset of neurodegenerative diseases” describes an experimental model to study the role of cytochrome p450 in neuronal processes involved in the development of neurodegenerative diseases. The authors show that the presence and regulation of cytochrome p450 isoenzymes in SH-SY5Y neuroblastoma cells justify the use of this model to study the involvement of drug metabolic systems in the therapeutic effects of drugs affecting the brain, as well as in neurodegeneration provoked by exogenous or endogenous compounds.

The paper is well presented and well written.

However, there are some comments on submitting the article.

Major

Figure 7. Dear authors! The results presented in the figure are compared from different blots. It is not correct. The results to be compared must be placed on the same blot. The results being compared must be presented on the same blot.

The number of repetitions should be increased to at least 4

Minor

1. line 38…Decipher cytochrome p450 again… cytochrome p450 (CYP)

2. line 44… …”in situ” replace with “in situ”

3. line 55… decipher MPP+

4. line 114 … “ß-tubulin” replace with “β-tubulin”

5. line 176… “2.2 and - and 5.9- fold” replace with “2.2- fold and 5.9- fold”

6. Reference 4 no DOI

7. There are many references older than 10 years; if it is possible to replace them with earlier ones, I ask the authors to do this.

Author Response

The article of Alice Pifferi et al. “Exploring the regulation of cytochrome P450 in SH-SY5Y cells: implications for the onset of neurodegenerative diseases” describes an experimental model to study the role of cytochrome p450 in neuronal processes involved in the development of neurodegenerative diseases. The authors show that the presence and regulation of cytochrome p450 isoenzymes in SH-SY5Y neuroblastoma cells justify the use of this model to study the involvement of drug metabolic systems in the therapeutic effects of drugs affecting the brain, as well as in neurodegeneration provoked by exogenous or endogenous compounds. The paper is well presented and well written. However, there are some comments on submitting the article.

Major

Figure 7. Dear authors! The results presented in the figure are compared from different blots. It is not correct. The results to be compared must be placed on the same blot. The results being compared must be presented on the same blot.

The number of repetitions should be increased to at least 4

We thank the Referee for raising this point. As we already reported in Figure 7 caption, due to the numerous experimental points, cells lysates of the same experiment were loaded in different gels/blots, run in parallel in the same chamber. We have however performed additional experiments by loading cell lysates (treated and relative controls) in the same gel and rearranged exemplificative blots and graphs differently in order to fulfil the request. Regarding the replicates, we have now reported in each figure legends the precise number, which is always higher than 4.

Minor

  1. line 38…Decipher cytochrome p450 again… cytochrome p450 (CYP).
  2. line 44… …”in situ” replace with “in situ”
  3. line 55… decipher MPP+
  4. line 114 … “ß-tubulin” replace with “β-tubulin”
  5. line 176… “2.2 and - and 5.9- fold” replace with “2.2- fold and 5.9- fold”
  6. Reference 4 no DOI

All the minor points 1-6 have been amended as requested. For reference 4, no DOI is available.

  1. There are many references older than 10 years; if it is possible to replace them with earlier ones, I ask the authors to do this.

When possible, old references have been replaced or deleted as requested, leaving however the old papers that historically represent the cornerstones of methodologies or famous basilar experimental observations.

Reviewer 2 Report

Comments and Suggestions for Authors There is a great lack of research in the field of degenerative processes in the brain in the face of neurological damage events in over 60% of the elderly population. Therefore, any research in the field that adds even a little knowledge is welcome even if you don't see an immediate application of the results. In the article you attached I have not yet seen the required graphics correction.    The aim of this study was to investigate the modulations of CYPs in neuronal adaptation. Changes that may affect the sensitivity of the brain to neurotoxins and it becomes clear that these toxins participate in the onset of neurodegenerative diseases. The study describes human individual differences in brain cytochrome P450 (CYP) metabolism. Although the research has implications for the third age - the experimental approach focused on differentiating SH-SY5Y human neuroblastoma cells (one of the most common solid tumors in children) to a phenotype similar to adult dopamine neurons. The study offers an experimental model to simulate the role of CYPs in neural processes involved in the development of neurodegenerative diseases and thus its contribution. The UD column in green is unnecessary because it is clear that the measurements are compared relative to it and it is against itself = 1. Therefore, it devalues the reader and lists unnecessary information.

Author Response

Ref #2

There is a great lack of research in the field of degenerative processes in the brain in the face of neurological damage events in over 60% of the elderly population. Therefore, any research in the field that adds even a little knowledge is welcome even if you don't see an immediate application of the results. In the article you attached I have not yet seen the required graphics correction.    The aim of this study was to investigate the modulations of CYPs in neuronal adaptation. Changes that may affect the sensitivity of the brain to neurotoxins and it becomes clear that these toxins participate in the onset of neurodegenerative diseases. The study describes human individual differences in brain cytochrome P450 (CYP) metabolism. Although the research has implications for the third age - the experimental approach focused on differentiating SH-SY5Y human neuroblastoma cells (one of the most common solid tumors in children) to a phenotype similar to adult dopamine neurons. The study offers an experimental model to simulate the role of CYPs in neural processes involved in the development of neurodegenerative diseases and thus its contribution. The UD column in green is unnecessary because it is clear that the measurements are compared relative to it and it is against itself = 1. Therefore, it devalues the reader and lists unnecessary information.

Thank you for your insightful comments. Your feedback on the graphical corrections has been noted, but considering that we had to rearrange the graphs according to the requests of other Referees to make them clearer for the reader, we believe that removing the normalization bar might create confusion, especially when comparing the basal levels in differentiated cells to those in undifferentiated cells.

Reviewer 3 Report

Comments and Suggestions for Authors

Minor Comments:

1.     The quantitative analysis for the expression of proteins Synaptophysin and ß-tubulin III in Fig. 2 (right hand side) do not correlate with the WB analysis (left hand side). For instance, band intensity of Synaptophysin with RA-BDNF is much higher than RA-TPA. Similarly, band intensity of ß-tubulin III with RA-TPA is higher than RA-BDNF. But quantitative analysis (right hand side) does not reflect the same. Authors need to explain this.

2.     Like Fig. 2, Fig.3 has the same issue with quantitative analysis. Authors need to explain this.

3.     Authors should mention a paragraph in the discussion section about how this study could be implemented to investigate the role of CYPs in neuronal processes involved in the development of neurodegenerative diseases.

Comments on the Quality of English Language

Minor improvement required.

Author Response

1.The quantitative analysis for the expression of proteins Synaptophysin and ß-tubulin III in Fig. 2 (right hand side) do not correlate with the WB analysis (left hand side). For instance, band intensity of Synaptophysin with RA-BDNF is much higher than RA-TPA. Similarly, band intensity of ß-tubulin III with RA-TPA is higher than RA-BDNF. But quantitative analysis (right hand side) does not reflect the same. Authors need to explain this.

  1. Like Fig. 2, Fig.3 has the same issue with quantitative analysis. Authors need to explain this.

We thank the Referee for raising these points. The intensity of the bands in the WB analysis has been properly calculated by normalizing to the intensity of beta-actin. For this reason, despite the band intensity of Synaptophysin with RA-BDNF seems much higher than RA-TPA, it should be compared/normalized to that of beta-actin in the same plot. However, we agree that in some cases, the representative images we selected may not properly reflect the quantitative analysis. Therefore, new blots have been performed, and misleading images have been replaced. The same applies to MAO enzymes in Figure 3.

  1. Authors should mention a paragraph in the discussion section about how this study could be implemented to investigate the role of CYPs in neuronal processes involved in the development of neurodegenerative diseases.

We thank the Referee for suggesting us to clarify this point. To fulfil this request, a new paragraph has now been inserted in the discussion.

Round 2

Reviewer 1 Report

Comments and Suggestions for Authors

I think the article should be published in this journal

Comments on the Quality of English Language

Punctuation should be checked